# BiDM: Pushing the Limit of Quantization for Diffusion Models

**Xingyu Zheng**[1], **Xianglong Liu**[✉1], **Yichen Bian**[1], **Xudong Ma**[1], **Yulun Zhang**[2],
**Jiakai Wang**[3], **Jinyang Guo**[1] , **Haotong Qin**[4]
[1]Beihang University    [2]Shanghai Jiao Tong University
[3]Zhongguancun Laboratory    [4]ETH Zürich
{zhengxingyu,xlliu,macaronlin,jinyangguo}@buaa.edu.cn
{yichen.bian.work,yulun100}@gmail.com wangjk@zgclab.edu.cn
haotong.qin@pbl.ee.ethz.ch

## Abstract

Diffusion models (DMs) have been significantly developed and widely used in various applications due to their excellent generative qualities. However, the expensive computation and massive parameters of DMs hinder their practical use in resource-constrained scenarios. As one of the effective compression approaches, quantization allows DMs to achieve storage saving and inference acceleration by reducing bit-width while maintaining generation performance. However, as the most extreme quantization form, 1-bit binarization causes the generation performance of DMs to face severe degradation or even collapse. This paper proposes a novel method, namely **BiDM**, for fully binarizing weights and activations of DMs, pushing quantization to the 1-bit limit. From a temporal perspective, we introduce the *Timestep-friendly Binary Structure* (TBS), which uses learnable activation binarizers and cross-timestep feature connections to address the highly timestep-correlated activation features of DMs. From a spatial perspective, we propose *Space Patched Distillation* (SPD) to address the difficulty of matching binary features during distillation, focusing on the spatial locality of image generation tasks and noise estimation networks. As the first work to fully binarize DMs, the W1A1 BiDM on the LDM-4 model for LSUN-Bedrooms 256×256 achieves a remarkable FID of 22.74, significantly outperforming the current state-of-the-art general binarization methods with an FID of 59.44 and invalid generative samples, and achieves up to excellent 28.0× storage and 52.7× OPs savings.

## 1 Introduction

Diffusion models (DMs) [19, 50, 44, 76], as a type of generative visual model [66, 59, 68], have garnered impressive attention and applications in various fields, such as image [57, 58], speech [42, 45, 24], and video [40, 18], because of their high-quality and diverse generative capabilities. The diffusion model can generate data from random noise through up to 1000 denoising steps [19]. Although some accelerated sampling methods effectively reduce the number of steps required for generating tasks [56, 31], the expensive floating-point computation of each timestep still limits its wide application on resource-constrained scenarios. Therefore, compression of the diffusion model becomes a crucial step for its broader application, and existing compression methods mainly include quantization [30, 54, 47], distillation [53, 36, 41, 73, 11], pruning [7, 12, 14, 13], *etc*. These compression approaches aim to reduce storage and computation while preserving accuracy.

Quantization is considered a highly effective model compression technique [70, 9, 64, 21, 10], which quantizes the weights and/or activations to low-bit integers or binaries for compact storage and

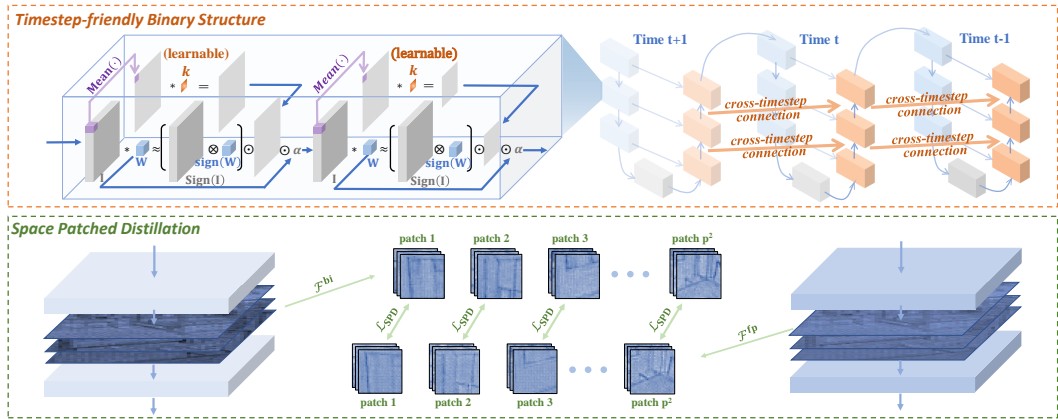

Figure 1: Overview of BiDM with *Timestep-friendly Binary Structure*, which improves DM architecture temporally, and *Space Patched Distillation*, which enhances DM optimization spatially.

efficient computation in inference. Some existing works thus apply quantization to compress DMs, aiming to compress and accelerate them while maintaining the quality of generation. Among them, 1-bit quantization, namely binarization, can achieve maximum storage savings for models and has performed well in discriminative models such as CNNs [33, 67, 65]. Furthermore, when both weights and activations are quantized to 1-bit, *e.g.*, fully binarized, efficient bitwise operations such as XNOR and bitcount can replace matrix multiplication, achieving the most efficient acceleration [74].

Some existing works have attempted to quantize DM to 1-bit [77], but their exploration mainly focuses on the weights, which are still far from full binarization. In fact, for generative models like DM, the impact of fully binarizing weights and activations is catastrophic: a) As generative models, DMs have rich intermediate representations closely related to timesteps and highly dynamic activation ranges, which are both very limited in information when binarized weights and activations are used; b) Generative models like DMs are typically required to output complete images, but the highly discrete parameter and feature space make it particularly difficult for binarized DMs to match the ground truth during training. The limited representational capacity, which is hard to match with timesteps dynamically, and the optimization difficulty of generative tasks in discrete space make it difficult for the binarized DM to converge or even collapse during the optimization process.

We propose **BiDM** to push diffusion models towards extreme compression and acceleration through complete binarization of weights and activations. It is designed to address the unique properties of DMs' activation features, model structure, and the demands of generative tasks, overcoming the difficulties associated with complete binarization. BiDM consists of two novel techniques: *From a temporal perspective*, we observe that the activation properties of DMs are highly correlated with timesteps. We introduce the Timestep-friendly Binary Structure (TBS), which uses learnable activation binary quantizers to match the highly dynamic activation ranges of DMs and designs feature connections across timesteps to leverage the similarity of features between adjacent timesteps, thereby enhancing the representation capacity of the binary model. *From a spatial perspective*, we note the spatial locality of DMs in generative tasks and the convolution-based U-Net structure. We propose Space Patched Distillation (SPD), which introduces a full-precision model as a supervisor and uses attention-guided imitation on divided patches to focus on local features, better guiding the optimization direction of the binary diffusion model.

Extensive experiments show that compared to existing SOTA fully binarized methods, BiDM significantly improves accuracy while maintaining the same inference efficiency, surpassing all existing baselines across various evaluation metrics. Specifically, in pixel space diffusion models, BiDM is the only method that raises the IS to 5.18, close to the level of full-precision models and 0.95 higher than the best baseline method. In LDM, BiDM reduces the FID on LSUN-Bedrooms from the SOTA method's 59.44 to an impressive 22.74, while fully benefiting from 28.0× storage and 52.7× OPs savings. As the first fully binarized method for diffusion models, numerous generated samples also demonstrate that BiDM is currently the only method capable of producing acceptable images with fully binarized DMs, enabling the efficient application of DMs in low-resource scenarios.

## 2 Related Work

Diffusion models (DMs) have demonstrated excellent generative capabilities across various tasks [19, 57, 58, 43, 42, 45, 24]. However, their large-scale model architectures and the high computational costs required for multi-step inference limit their practical applications. To address this, methods for accelerating the process at the timestep level have been widely proposed, including sampling acceleration that does not require retraining [56, 31, 34, 35] and distillation methods [53, 36, 41]. A recent method called DeepCache [38] caches high-dimensional features to avoid a lot of redundant computations and is compatible with typical sampling acceleration methods. However, these methods cannot overcome the memory bottlenecks and efficiency limits during single-step inference.

Quantization is a widely validated compression technique that compresses weights and activations from the usual 32 bits to 1-8 bits to achieve compression and acceleration [6, 78, 37, 75]. Consequently, quantization is being studied for application in diffusion models [15, 4]. These methods generally consider the unique timestep structure and spatial architecture of diffusion models, but due to the significant difficulty of quantizing generative models, most post-training quantization (PTQ) methods can only quantize models to 4 bits or more [29, 54, 22], while more accurate quantization-aware training (QAT) methods face severe performance bottlenecks below 3 bits [30, 55].

Binarization, the most extreme form of quantization, typically expresses weights and activations as ±1, allowing the model to achieve maximum compression and acceleration [60, 62]. In computer vision, binarization work has mainly focused on discriminative models like CNNs [49, 33, 46, 48] or ViTs [28, 16], with limited work on generative models. While ResNet VAE and Flow++ [1] have achieved complete binarization for VAEs [26], they do not offer generative performance comparable to current advanced models. Binary Latent Diffusion [61] binarized the latent space of LDMs [26] but did not improve the model's spatial footprint or inference efficiency. The latest work, BinaryDM [50], quantized DMs to nearly W1A4, but it did not address activation quantization, leaving room for achieving full binarization and acceleration of DMs.

## 3 Method

### 3.1 Binarized Diffusion Model Baseline

**Diffusion models.** Given a data distribution $x_0 \sim q(x_0)$, the forward process generates a sequence of random variables $x_t \in \{x_1, \cdots, x_T\}$ with transition kernel $q(x_t|x_{t-1})$, usually Gaussian perturbation, which can be expressed as

$$q\left(x_1, \ldots, x_T \mid x_0\right) = \prod_{t=1}^{T} q(x_t \mid x_{t-1}), \quad q\left(x_t \mid x_{t-1}\right) = \mathcal{N}\left(x_t; \sqrt{1-\beta_t}x_{t-1}, \beta_t I\right), \quad (1)$$

where $\beta_t \in (0,1)$ is a noise schedule. Gaussian transition kernel allows us to marginalize the joint distribution, so with $\alpha_t := 1 - \beta_t$ and $\bar{\alpha}_t := \prod_{i=1}^{t} \alpha_i$, we can easily obtain a sample of $x_t$ by sampling a gaussian vector $\epsilon \sim \mathcal{N}(0, I)$ and applying the transformation $x_t = \sqrt{\bar{\alpha}_t}x_0 + \sqrt{1 - \bar{\alpha}_t}\epsilon$.

The reverse process aims to generate samples by removing noise, approximating the unavailable conditional distribution $q\left(x_{t-1} \mid x_t\right)$ with a learnable transition kernel $p_\theta\left(x_{t-1} \mid x_t\right)$, which can be expressed as

$$p_\theta\left(x_{t-1} \mid x_t\right) = \mathcal{N}\left(x_{t-1}; \tilde{\mu}_\theta\left(x_t, t\right), \tilde{\beta}_t I\right). \quad (2)$$

The mean $\tilde{\mu}_\theta\left(x_t, t\right)$ and variance $\tilde{\beta}_t$ could be derived using the reparameterization tricks in [19]:

$$\tilde{\mu}_\theta\left(x_t, t\right) = \frac{1}{\sqrt{\alpha_t}}\left(x_t - \frac{1-\alpha_t}{\sqrt{1-\bar{\alpha}_t}}\epsilon_\theta\left(x_t, t\right)\right), \quad \tilde{\beta}_t = \frac{1-\bar{\alpha}_{t-1}}{1-\bar{\alpha}_t} \cdot \beta_t, \quad (3)$$

where $\epsilon_\theta$ is a function approximation with the learnable parameter $\theta$, which predicts $\epsilon$ given $x_t$.

For the training of DMs, a simplified variant of the variational lower bound is usually applied as the loss function for better sample quality, which can be expressed as

$$\mathcal{L}_{\text{DM}} = \mathbb{E}_{t \sim [1,T], x_0 \sim q(x_0), \epsilon \sim \mathcal{N}(0,I)}\left[\left\|\epsilon_t - \epsilon_\theta\left(x_t, t\right)\right\|^2\right]. \quad (4)$$

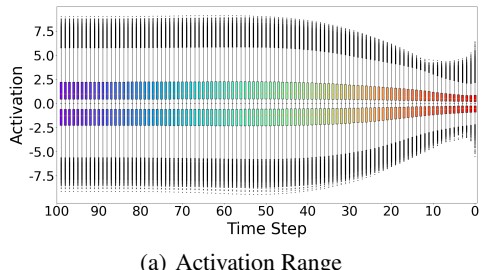

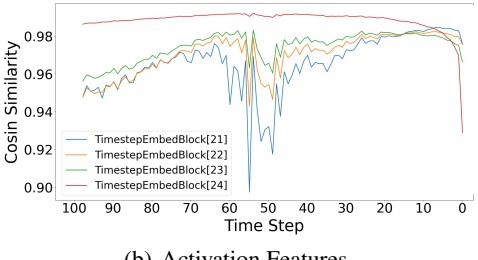

| (a) Activation Range | (b) Activation Features |

Figure 2: (a) The activation range of the 4th convolutional layer of the full-precision DDIM model on CIFAR-10 varies with the denoising timesteps. (b) The output features are similar at each step of the full-precision LDM-4 model on LSUN-Bedrooms compared to the previous step.

U-Net [51], due to its ability to fuse low-level and high-dimensional features, has become the mainstream backbone of Diffusion. The input-output blocks of U-Net can be represented as $\{D_m\}_{m=1}^d$ and $\{U_m\}_{m=1}^d$, where blocks corresponding to smaller $m$ are more low-level. Skip connections propagate low-level information from $D_m(\cdot)$ to $U_m(\cdot)$, so the input received by $U_m$ is expressed as:

$$\text{Concat}(D_m(\cdot), U_{m+1}(\cdot)). \tag{5}$$

**Binarization.** The quantization compresses and accelerates the noise estimation model by discretizing weights and activations to low bit-width. In the baseline of the binarized diffusion model, the weights $\boldsymbol{w}$ are binarized to 1-bit [49, 5, 20]:

$$\boldsymbol{w}^{\text{bi}} = \sigma \, \text{sign}(\boldsymbol{w}) = \begin{cases} \sigma, & \text{if } \boldsymbol{w} \geq 0, \\ -\sigma, & \text{otherwise,} \end{cases} \tag{6}$$

where $\text{sign}$ function confine $\boldsymbol{w}$ to +1 or -1 with 0 thresholds. $\sigma$ is a floating-point scalar, which is initialized as $\frac{\|\boldsymbol{w}\|}{n}$ ($n$ denotes the number of weights) and learnable during training following [49, 33].

Meanwhile, activations are typically quantized by naive BNN quantizers [23, 32]:

$$\boldsymbol{a}^{\text{bi}} = \text{sign}(\boldsymbol{a}) = \begin{cases} 1, & \text{if } \boldsymbol{a} \geq 0, \\ -1, & \text{otherwise.} \end{cases} \tag{7}$$

When both weights and activations are quantized to 1-bit, the computations of the denoising model can be replaced by XNOR and bitcount operators, achieving significant compression and acceleration.

## 3.2 Timestep-friendly Binary Structure

Before delving into the detailed description of the proposed method, we summarize our observation on the properties of DMs:

**Observation 1.** *The activation range varies significantly across long-term timesteps, but the activation features are similar in short-term neighbouring timesteps.*

Previous works, such as TDQ [55] and Q-DM [30], have commonly demonstrated that the activation distribution of DMs largely depends on denoising process, manifesting as similarities between adjacent timesteps while difference between distant ones, as shown in Figure 2(a). Therefore, applying a fixed scaling factor to activations across all timesteps can cause significant distortion in the activation range. Beyond the distribution range, Deepcache [38] highlights the substantial temporal consistency of high-dimensional features across consecutive timesteps, as shown in Figure 2(b).

These phenomena prompt us to reexamine existing binary structures. Binaryization, especially the full binaryization of weights and activations, results in a greater loss of activation range and precision compared to low-bit quantizations like 4-bit [50]. This makes it more challenging to generate rich activation features. Such deficiencies in activation range and output features significantly harm representation-rich generative models like DMs. Therefore, adopting binary quantizers with more

flexible activation ranges for DMs, and enhancing the model's overall expressive power by leveraging its feature outputs, are crucial strategies for improving its generative capability after full binaryization.

We first focus on the differences between various timesteps over the long term. Most existing activation quantizers, such as BNN [23] and Bi-Real [32], as shown in Eq. (7), directly quantize activations to {+1, -1}. This approach significantly disrupts activation features and negatively impacts the expressive power of generative models. Some improved activation binary quantizers, such as XNOR++ [2], adopt a trainable scale factor $k$:

$$\boldsymbol{a}^{\text{bi}} = K \operatorname{sign}(\boldsymbol{a}) = \begin{cases} K, & \text{if } \boldsymbol{a} \geq 0, \\ -K, & \text{otherwise,} \end{cases} \tag{8}$$

where the form of $K$ could be either a vector or the product of multiple vectors, but it remains a constant value during inference. Although this approach partially restores the feature expression of activations, it does not align well with diffusion models that are highly correlated with timesteps and may still lead to significant performance loss.

We turn our attention to the original XNOR, which employs dynamically computed means to construct the activation binary quantizer. Its operation for 2D convolution can be expressed as:

$$\mathbf{I} * \mathbf{W} \approx (\operatorname{sign}(\mathbf{I}) \otimes \operatorname{sign}(\mathbf{W})) \odot (K\alpha) = (\operatorname{sign}(\mathbf{I}) \otimes \operatorname{sign}(\mathbf{W})) \odot (A * k\alpha), \tag{9}$$

where $\mathbf{I} \in \mathbb{R}^{c \times w_{in} \times h_{in}}$, $\mathbf{W} \in \mathbb{R}^{c \times w \times h}$, $A = \frac{\sum |\mathbf{I}_{i,:,:}|}{c}$, $\alpha = \frac{1}{n} \|\mathbf{W}\|_{\ell 1}$. $k \in \mathbb{R}^{1 \times 1 \times w \times h}$ represents a 2D filter, where $\forall ij \ k_{ij} = \frac{1}{w \times h}$. $*$ and $\otimes$ indicate convolution with and without multiplication, respectively. This approach naturally preserves the range of activation features and dynamically adapts with the input range across different timesteps. However, due to the rich expression of DM features, local activations exhibit inconsistency in range before and after passing through modules, indicating that the predetermined value of $k$ does not effectively restore the activation representation.

Therefore, we make $k$ adjustable and allow it to be learned during training to adaptively match the changes in the range of activations before and after. The gradient calculation process of our learnable tiny convolution $k$ can be expressed as follows:

$$\frac{\partial \mathcal{L}}{\partial k} = \frac{\partial \mathcal{L}}{\partial (\mathbf{I} * \mathbf{W})} \frac{\partial (A * k\alpha)}{\partial k} (\operatorname{sign}(\mathbf{I}) \otimes \operatorname{sign}(\mathbf{W})). \tag{10}$$

Notably, making $k$ learnable does not add any extra inference burden. The computational cost remains unchanged, allowing for efficient binary operations.

On the other hand, we focus on the similarity between adjacent timesteps. Deepcache directly extracts high-dimensional features as a cache to skip a large amount of deep computation in U-Net, achieving significant inference acceleration. This process is expressed as:

$$F_{cache}^t \leftarrow U_{m+1}^t(\cdot), \quad \operatorname{Concat}(D_m^{t-1}(\cdot), F_{cache}^t). \tag{11}$$

However, this approach does not apply to binarized diffusion models, as the information content of each output from a binary network is very limited. For binary diffusion models, which inherently achieve significant compression and acceleration but have limited expressive power, we anticipate that the similarity of features between adjacent timesteps will enhance binary representation, thereby compensating for the representation challenges.

We construct a cross-timestep information enhancement connection to enrich the expression at the current timestep using features from the previous step. This process can be expressed as:

$$\operatorname{Concat}(D_m^{t-1}(\cdot), (1 - \alpha_{m+1}^{t-1}) \cdot U_{m+1}^{t-1}(\cdot) + \alpha_{m+1}^{t-1} \cdot U_{m+1}^t(\cdot)), \tag{12}$$

where $\alpha_{m+1}^{t-1}$ is a learnable scaling factor. As shown in Figure 2(b), the similarity of high-dimensional features varies across different blocks and timesteps in DMs. Therefore, we set multiple independent $\alpha$ values to allow the model to adaptively learn more effectively during training.

In summary, Timestep-friendly Binary Structure (TBS) includes learnable tiny convolution applied to scaling factors after averaging the inputs and connections across timesteps. Their combined effect adapts to the changes in the activation range of diffusion models over long-range timesteps and leverages the similarity of high-dimensional features between adjacent timesteps to enhance information representation.

From the perspective of error reduction, a visualization of TBS is shown in Figure 3. First, we abstract the output of the binary DM under the baseline method as vector $B^{t-1}$. The mismatch in scaling factors creates a significant difference in length between it and the output vector $F^{t-1}$ of the full-precision model. Using our proposed scaling factors and learnable tiny convolutions, $B^{t-1}$ is expanded to $L^{t-1}$. $L^{t-1}$ is closer to $F^{t-1}$, but there is still a directional difference from the full-precision model. The cross-timestep connection further incorporates the outputs $F^t$ of the previous timestep, $B^t$, and $L^t$. The high-dimensional feature similarity between adjacent timesteps means the gap between $F^{t-1}$ and $F^t$

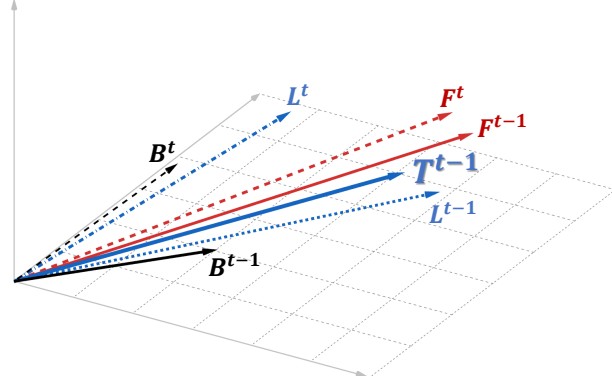

Figure 3: An illustration of TBS. Since the feature space is high-dimensional, we illustrate it using schematic diagrams.

is relatively small, facilitating the combination of $L^{t-1}$ and $L^t$. Finally, we obtain the binarized DM's output with TBS applied as $T^{t-1} = (1 - \alpha) \cdot L^{t-1} + \alpha \cdot L^t$, closest to the output $F^{t-1}$ of the full-precision model. The learnable tiny convolution $k$ in TBS allows scaling factors to adapt more flexibly to the representation of DM, while connections across timesteps enable the binarized DM to use the previous step's output information for appropriate information compensation.

### 3.3 Space Patched Distillation

Due to the nature of generative models, the optimization process of diffusion models exhibits different characteristics from past discriminative models:

**Observation 2.** *Conventional distillation struggles to guide fully binarized DMs to align with full-precision DMs, while the features of DM exhibit locality in space during the generation task.*

In previous practices, adding distillation loss during the training of quantized models has been a common approach. As the numerical space of binary models is limited, directly optimizing them using naive loss leads to difficulties in adjusting gradient update directions and makes learning challenging. Therefore, adding distillation loss to intermediate features can better guide the model's local and global optimization process.

However, as a generative model, the highly rich feature representation of DMs makes it extremely difficult for binary models to finely mimic full-precision models. Although the L2 loss used in the original DM training aligns with the Gaussian noise in the diffusion process, it is not suitable for the distillation matching of intermediate features. During regular distillation, the commonly used L2 loss tends to prioritize optimizing pixels with larger discrepancies, leading to a more uniform and smooth optimization result. This global constraint learning process is challenging for binary models aimed at image generation, as their limited representation capacity makes it difficult for fine-grained distillation imitation to directly adjust them to fully match the direction of full-precision models.

At the same time, we note that DMs using U-Net as a backbone naturally exhibit spatial locality due to their convolution-based structure and generative task requirements. This is different from past discriminative models, where tasks like classification only require overall feature extraction without low-level requirements, making traditional distillation methods unsuitable for DMs with spatial locality in generative tasks. Additionally, most existing DM distillation methods focus on reducing the number of timesteps and do not address the spatial locality of features required for image generation tasks.

Therefore, given the difficulty in optimizing binary DMs with existing loss functions and the spatial locality of DMs, we propose Space Patched Distillation (SPD). Specifically, we designed a new loss function that partitions features into patches before distillation and then calculates spatial attention-guided loss patch by patch. While conventional L2 loss makes it difficult for binary DMs to achieve direct matching, leading to optimization challenges, the attention mechanism allows the distillation

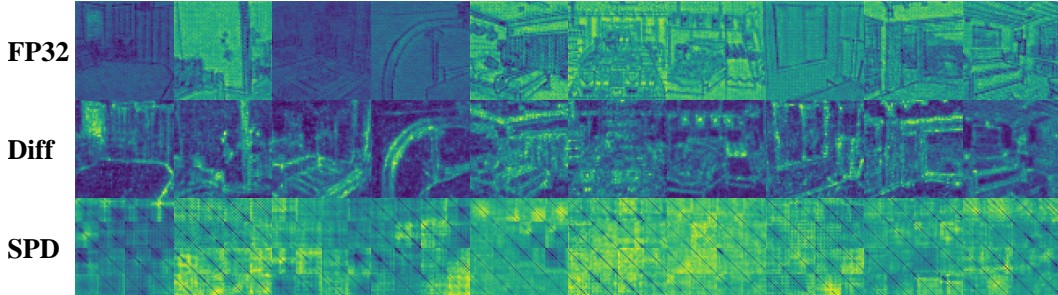

Figure 4: Visualization of the last TimeStepBlock's output of the LDM model on LSUN-bedroom dataset. FP32 denotes the full-precision model's output $\mathcal{F}^{\text{fp}}$. Diff denotes the difference between the output of the full-precision model and the binarized one $\left\|\mathcal{F}^{\text{fp}} - \mathcal{F}^{\text{bi}}\right\|$. Ours denotes the attention-guided SPD.

optimization to focus more on critical parts. However, this is still challenging for fully binarized DMs because the highly discrete binary outputs have limited information, making it difficult for the model to capture global information. Therefore, we leverage the spatial locality of DMs by dividing intermediate features into multiple patches and independently calculating spatial attention-guided loss for each patch, allowing the binary model to better utilize local information during optimization.

SPD first divides the intermediate features $\mathcal{F}^{\text{bi}}$ and $\mathcal{F}^{\text{fp}} \in \mathbb{F}^{b \times c \times w \times h}$, output by a block of the binary DM and the full-precision DM respectively, into $p^2$ patches:

$$\mathcal{P}^{\text{fp}}_{i,j} = \mathcal{F}^{\text{fp}}_{[:,:,i:i+w/p,j:j+h/p]}, \quad \mathcal{P}^{\text{bi}}_{i,j} = \mathcal{F}^{\text{bi}}_{[:,:,i:i+w/p,j:j+h/p]}. \tag{13}$$

Then, attention-guided loss is calculated for each patch separately:

$$\mathcal{A}^{\text{fp}}_{i,j} = \mathcal{P}^{\text{fp}}_{i,j} \mathcal{P}^{\text{fp}}_{i,j}{}^{T}, \quad \mathcal{A}^{\text{bi}}_{i,j} = \mathcal{P}^{\text{bi}}_{i,j} \mathcal{P}^{\text{bi}}_{i,j}{}^{T}. \tag{14}$$

After regularization, the losses at corresponding positions are calculated and summed up:

$$\mathcal{L}^{m}_{\text{SPD}} = \frac{1}{p^2} \sum_{i=0}^{p-1} \sum_{j=0}^{p-1} \left\| \frac{\mathcal{A}^{\text{fp}}_{i,j}}{\|\mathcal{A}^{\text{fp}}_{i,j}\|_2} - \frac{\mathcal{A}^{\text{bi}}_{i,j}}{\|\mathcal{A}^{\text{bi}}_{i,j}\|_2} \right\|_2, \tag{15}$$

where $\|\cdot\|_2$ denotes the L2 function. Finally, the total training loss $\mathcal{L}$ is computed as:

$$\mathcal{L} = \mathcal{L}_{\text{DM}} + \frac{\lambda}{2d+1} \sum_{m}^{2d+1} \mathcal{L}^{m}_{\text{SPD}}, \tag{16}$$

where $d$ denotes the number of blocks during the upsampling process or downsampling process, resulting in a total of $2d+1$ intermediate features, including the middle block. $\lambda$ is a hyperparameter coefficient to balance the loss terms, defaulting set to 4.

We visualize the intermediate features and attention-guided SPD mentioned above. As Figure 4 shown, our SPD allows the model to pay more attention to local information in each patch.

## 4 Experiment

We conduct experiments on various datasets, including CIFAR-10 $32 \times 32$ [27], LSUN-Bedrooms $256 \times 256$ [72], LSUN-Churches $256 \times 256$ [72] and FFHQ $256 \times 256$ [25] over pixel space diffusion models [19] and latent space diffusion models [50]. The evaluation metrics used in our study encompass Inception Score (IS), Fréchet Inception Distance (FID) [17], Sliding Fréchet Inception Distance (sFID) [52], Precision and Recall. To date, there has been no research that compresses diffusion models to such an extreme extent. Therefore, we use classical binarization algorithms [2, 78, 33, 49], the recent SOTA general binarization algorithms [62], and quantization methods suited to generative models [15, 63] as baselines. We extract the outputs of TimestepEmbedBlocks from the DM to serve as the operating target for our TBS and SPD. And we employ the same shortcut connections in convolutional layers as those used in ReActNet[33]. Detailed experiment settings are presented in the Appendix A.

## 4.1 Main Results

**Pixel Space Diffusion Models.** We first conduct experiments on the CIFAR-10 $32 \times 32$ dataset. As the results presented in Table 1, W1A1 binarization of DM using baseline methods results in substantial degradation. However, BiDM demonstrated significant improvements across all metrics, achieving unprecedented restoration of image quality. Specifically, BiDM achieved remarkable enhancements from 4.23 to 5.18 in the IS metric, and reduced 27.9% in the FID metric.

Table 1: Binarization results for DDIM on CIFAR-10 datasets with 100 steps.

| Model | Dataset | Method | #Bits | IS↑ | FID↓ | sFID↓ | Precision↑ |
|-------|---------|--------|-------|-----|------|-------|------------|
| DDIM | CIFAR-10 $32 \times 32$ | FP | 32/32 | 8.90 | 5.54 | 4.46 | 67.92 |
| | | XNOR++[2] | 1/1 | 2.23 | 251.14 | 60.85 | 44.98 |
| | | DoReFa[78] | 1/1 | 1.43 | 397.60 | 139.97 | 0.17 |
| | | ReActNet[33] | 1/1 | 3.35 | 231.55 | 119.80 | 18.37 |
| | | ReSTE[62] | 1/1 | 1.26 | 394.29 | 125.84 | 0.18 |
| | | XNOR[49] | 1/1 | 4.23 | 113.36 | 27.67 | 46.96 |
| | | **BiDM** | **1/1** | **5.18** | **81.65** | **25.68** | **52.92** |

**Latent Space Diffusion Models.** Our LDM experiments encompass the evaluation of LDM-4 on LSUN-Bedrooms $256 \times 256$ and FFHQ $256 \times 256$ datasets, along with the assessment of LDM-8 on the LSUN-Churches $256 \times 256$ dataset. The experiments utilized the DDIM sampler with 200 steps, and the detailed outcomes are presented in Table 2. Across these three datasets, our method achieved significant improvements over the best baseline methods. In comparison to other binarization algorithms, BiDM outperformed across all metrics. On the LSUN-Bedrooms, LSUN-Churches, and FFHQ datasets, the FID metric of BiDM decreased by 61.7%, 30.7%, and 51.4%, respectively, compared to the best results among the baselines.

In contrast to XNOR++, its adoption of fixed activation scaling factors in the denoising process results in a very limited dynamic range for its activations, making it difficult to match the highly flexible generative representations of DMs. BiDM addressed this challenge by making the tiny convolution $k$ learnable, which acts on the dynamically computed scaling factors. This optimization led to substantial improvements exceeding an order of magnitude across all metrics. On the LSUN-Bedrooms and LSUN-Churches datasets, the FID metric decreased from 319.66 to 22.74 and from 292.48 to 29.70, respectively. Additionally, compared to the SOTA binarization method ReSTE, BiDM achieved significant enhancements across multiple metrics, particularly demonstrating notable improvements on the LSUN-Bedrooms dataset. We have supplemented our work with BBCU, a binarization method more akin to generative models like DMs rather than discriminative models. Experimental results indicate that even as a binarization strategy for generative models, BBCU faces significant breakdowns when applied to DMs, as FID dropped dramatically to 236.07 on LSUN-Bedrooms. As a work targeting QAT for DM, EfficientDM is indeed a suitable comparison, especially since it designs TALSQ to address the variation in activation range. The results show that EfficientDM struggles to adapt to the extreme scenario of W1A1, and this may be due to its quantizer having difficulty adapting to binarized DM, and using QALoRA for weight updates might yield suboptimal results compared to full-parameter QAT.

As we mentioned in the TBS section of our manuscript, most existing binarization methods struggle to handle the wide activation range and flexible expression of DMs, further highlighting the necessity of TBS. Their optimization strategies may also not be tailored for the image generation tasks performed by DM, which means they only achieve conventional but suboptimal optimization.

## 4.2 Ablation Study

We perform comprehensive ablation studies for LDM-4 on the LSUN-Bedrooms $256 \times 256$ dataset to evaluate the effectiveness of each proposed component in BiDM. We evaluate the effectiveness of our proposed SPD and TBS, and the results are presented in Table 3. Upon separately applying our SPD or TBS methods to LDM, we observed significant improvements compared to the original performance. When the TBS method was incorporated, FID and sFID dropped sharply from 106.62 and 56.61 to 35.23 and 25.13, respectively. Similarly, when the SPD method was added, FID and sFID decreased

Table 2: Quantization results for LDM on LSUN-Bedrooms, LSUN-Churches and FFHQ datasets.

| Model | Dataset | Method | #Bits | FID↓ | sFID↓ | Precision↑ | Recall↑ |
|---|---|---|---|---|---|---|---|
| LDM-4 | LSUN-Bedrooms 256 × 256 | FP | 32/32 | 2.99 | 7.08 | 65.02 | 47.54 |
| | | XNOR++ | 1/1 | 319.66 | 184.75 | 0.00 | 0.00 |
| | | BBCU | 1/1 | 236.07 | 89.66 | 0.59 | 5.66 |
| | | EfficientDM | 1/1 | 194.45 | 113.24 | 0.99 | 9.20 |
| | | DoReFa | 1/1 | 188.30 | 89.28 | 0.86 | 0.18 |
| | | ReActNet | 1/1 | 154.74 | 61.50 | 4.63 | 9.30 |
| | | ReSTE | 1/1 | 59.44 | 42.16 | 12.06 | 2.92 |
| | | XNOR | 1/1 | 106.62 | 56.81 | 6.82 | 5.22 |
| | | **BiDM** | 1/1 | **22.74** | **17.91** | **33.54** | **19.90** |
| LDM-8 | LSUN-Churches 256 × 256 | FP | 32/32 | 4.36 | 16.00 | 74.64 | 48.98 |
| | | XNOR++ | 1/1 | 292.48 | 168.65 | 0.02 | 0.00 |
| | | DoReFa | 1/1 | 162.06 | 95.37 | 7.85 | 0.74 |
| | | ReActNet | 1/1 | 56.39 | 54.68 | 45.13 | 2.06 |
| | | ReSTE | 1/1 | 47.88 | 52.44 | 51.98 | 3.34 |
| | | XNOR | 1/1 | 42.87 | 49.24 | 51.53 | 4.28 |
| | | **BiDM** | 1/1 | **29.70** | **45.14** | **55.75** | **14.80** |
| LDM-4 | FFHQ 256 × 256 | FP | 32/32 | 4.87 | 6.96 | 74.73 | 50.57 |
| | | XNOR++ | 1/1 | 379.49 | 320.64 | 0.00 | 0.00 |
| | | DoReFa | 1/1 | 214.06 | 177.63 | 2.09 | 0.00 |
| | | ReActNet | 1/1 | 147.88 | 141.31 | 3.36 | 0.69 |
| | | ReSTE | 1/1 | 144.37 | 97.43 | 4.03 | 0.03 |
| | | XNOR | 1/1 | 89.37 | 54.04 | 31.31 | 4.11 |
| | | **BiDM** | 1/1 | **43.42** | **32.35** | **49.44** | **13.96** |

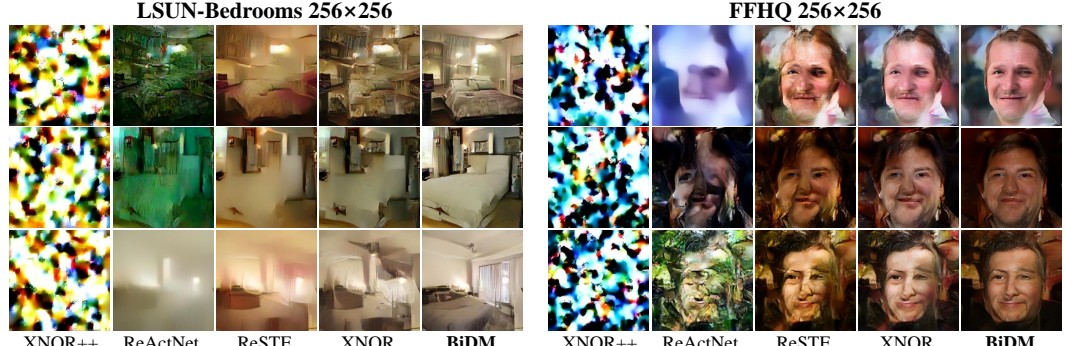

**LSUN-Bedrooms 256×256**   **FFHQ 256×256**

XNOR++  ReActNet  ReSTE  XNOR  **BiDM**     XNOR++  ReActNet  ReSTE  XNOR  **BiDM**

Figure 5: Visualization of samples generated by the W1A1 baseline and our BiDM. BiDM is the first fully binarized DM method capable of generating viewable images, significantly surpassing advanced binarization methods.

significantly from 106.62 and 56.61 to 40.62 and 31.61, respectively. Other metrics also exhibited substantial improvements. This demonstrates the effectiveness of our approach in continuously approximating the binarized model features to full-precision features during training by introducing a learnable factor $\alpha_m^t$ and incorporating connections between adjacent time steps. Furthermore, when we combined our two methods and applied them to LDM, we observed an additional improvement compared to the individual application of each method. This further substantiates that performing distillation between full-precision and binarized models at the patch level can significantly enhance the performance of the binarized model. We also conducted additional ablation experiments, and the results are presented in the appendix B.

Table 3: Ablation result of each proposed component.

| Method | #Bits | FID↓ | sFID↓ | Prec.↑ | Recall↑ |
|--------|-------|------|-------|--------|---------|
| Vanilla | 1/1 | 106.62 | 56.81 | 6.82 | 5.22 |
| +TBS | 1/1 | 35.23 | 25.13 | 26.38 | 14.32 |
| +SPD | 1/1 | 40.62 | 31.61 | 23.87 | 11.18 |
| **BiDM** | 1/1 | **22.74** | **17.91** | **33.54** | **19.90** |

## 4.3 Efficiency Analysis

**Inference Efficiency Analysis.** We conducted an analysis of the diffusion model's inference efficiency under complete binarization. During inference, BiDM requires only a very small number of additional floating-point additions for the connections across timesteps compared to the classic binarization work XNOR-Net, and there are no differences in the majority of calculations, such as convolutions. Performing a floating-point convolution with a depth of 1 for scaling factors requires only a small amount of computation, and the overhead for averaging matrix $A$ is also minimal. The findings presented in Table 4 reveal that BiDM, while achieving the same $28.0\times$ memory efficiency and $52.7\times$ computational savings as the XNOR baseline, demonstrates significantly superior image generation capabilities, with the FID decreased from 106.62 to 22.74. See Appendix B for more details.

Table 4: Inference efficiency of our proposed BiDM of LDM-4 on LSUN-Bedrooms.

| Method | #Bits | Size(MB) | BOPs($\times10^9$) | FLOPs($\times10^9$) | OPs($\times10^9$) | FID↓ |
|--------|-------|----------|--------------------|---------------------|-------------------|------|
| FP | 32/32 | 1045.4 | - | 96.00 | 96.00 | 2.99 |
| XNOR | 1/1 | 37.3 | 92.1 | 0.38 | 1.82 | 106.62 |
| **BiDM** | 1/1 | 37.3 | 92.1 | 0.38 | 1.82 | 22.74 |

**Training Efficiency Analysis.** We also explored the training efficiency of BiDM, as the overhead required for the QAT of binarized DMs cannot be overlooked. Theoretical analysis and experimental results show that BiDM achieved significantly better generative results than baseline methods under the same training cost, demonstrating that it not only has a higher upper limit of generative capability but is also relatively efficient in terms of generative performance. See Appendix B for details.

**Limitations.** The techniques of BiDM increase the training time of DMs compared with the original process, and future efforts may thus focus on the efficient quantization process of DMs.

## 5 Conclusion.

In this paper, we present BiDM, a novel fully binarized method that pushes the compression of diffusion models to the limit. Based on two observations — activations at different timesteps and the characteristics of image generation tasks — we propose the Timestep-friendly Binary Structure (TBS) and Space Patched Distillation (SPD) from temporal and spatial perspectives, respectively. These methods address the severe limitations in representation capacity and the challenges of highly discrete spatial optimization in full binarization. As the first fully binarized diffusion model, BiDM demonstrates significantly better generative performance than the SOTA general binarization methods across multiple models and datasets. On LSUN-Bedrooms, BiDM achieves an FID of 22.74, greatly surpassing the SOTA method with an FID of 59.44, making it the only method capable of generating visually acceptable samples while achieving up to $28.0\times$ storage savings and $52.7\times$ OPs savings.

## Acknowledgments and Disclosure of Funding

This work was supported by the Beijing Municipal Science and Technology Project (No. Z231100010323002), the National Natural Science Foundation of China (Nos. 62306025, 92367204), and the Fundamental Research Funds for the Central Universities.

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

# A  Experiment Settings

We adopt several classic binarization algorithms, including XNOR [49], XNOR++ [2], DoReFa [78], and ReActNet [33], along with the SOTA binarization method, ReSTE [62] as baselines. Additionally, we also include the quantization methods designed for generative models, BBCU [63] and EfficientDM [15]. We extract the output features of TimestepEmbedBlocks from the DM to serve as the targets of TBS and SPD operations. For the CIFAR-10 [27] dataset, We add TBS connections to the outputs of the last 2 timestep embedding blocks and set $\alpha_{init}$ to 0.3. The $\lambda$ on CIFAR-10 is set to 3e-2. For the LSUN-Bedrooms [72], LSUN-Churches [72] and FFHQ [25] datasets, We add TBS connections to the outputs of the last 8 timestep embedding blocks and also set $\alpha_{init}$ to 0.3. The $\lambda$ on these three datasets is set to 1e-2.

Our quantization-aware training is based on the pre-trained diffusion model, and the quantizer parameters and latent weights are trained simultaneously. The overall training process is relatively consistent with the original training process of DDIM or LDM. For the CIFAR-10 dataset, we set the learning rate to 6e-5 and the batch size to 64 during training. The training process consisted of 100k iterations, and during sampling, we used 100 sampling steps. For the LSUN-Bedrooms, LSUN-Churches and FFHQ datasets, the learning rate was set to 2e-5 and the batch size to 4 during training. The training consisted of 200k iterations, with 200 steps used during denoising phase.

We conducted extensive experiments on two different types of diffusion models: the latent-space diffusion model LDM and the pixel-space diffusion model DDIM. For the DDIM model, we specifically selected the CIFAR-10 dataset with a resolution of $32 \times 32$ for our experiments. For the LDM model, our experiments spanned multiple datasets, including the LSUN-Bedrooms, LSUN-Churches and the FFHQ dataset, all with a resolution of $256 \times 256$. To evaluate the generation quality of the diffusion model, we utilize several evaluation metrics, including Inception Score (IS), Fréchet Inception Distance (FID) [17], Sliding Fréchet Inception Distance (sFID) [52], and Precision-and-Recall. After 200,000 iterations of training, we randomly sample and generate 50,000 images from the model and compute the metrics based on reference batches. The reference batches used to evaluate FID and sFID contain all the corresponding datasets. We recorded FID, sFID, and Precision for all tasks and additional IS for CIFAR-10.

We utilize OPs as metrics for evaluating theoretical inference efficiency. Taking the convolutional unit as an example, the BOPs for a single computation operation of a single convolution are defined as follows $nmk^2b_ab_w$ [69, 71]. It is composed of $b_w$ bits for weights, $b_a$ bits for activation, $n$ input channels, $m$ output channels, and a $k \times k$ convolutional kernel. For the output feature with width $w$ and height $h$, $BOPs \approx whnmk^2b_ab_w$. As there might also be full-precision modules in the model, the total OPs of the model are summed up as $\frac{1}{64}BOPs + FLOPs$ [3]. All our experiments are conducted on a server with NVIDIA A100 40GB GPU.

# B  Additional Quantitative Results

We conducted more detailed ablation experiments to comprehensively validate our results.

**Effects of learnable $k$ in TBS.** We apply the proposed learnable $k$ to the XNOR baseline. The experimental results shown in Table 5 indicate that this modification can lead to a significant improvement in performance. The model achieved a doubling of improvement in FID, sFID. Their original values were 106.62 and 56.81, respectively, and they decreased to 57.62 and 30.46. The negligible degradation in Recall can be overlooked.

Table 5: Solely transforming $k$ into learnable on the XNOR baseline network.

| $k$ | #Bits | FID↓ | sFID↓ | Prec.↑ | Recall↑ |
|---|---|---|---|---|---|
| Vanilla | 1/1 | 106.62 | 56.81 | 6.82 | 5.22 |
| learnable | 1/1 | 57.26 | 30.46 | 15.88 | 5.00 |

**Effects of cross-timestep connection in TBS.** We investigated the impact of varying the number of TBS connections. Table 6 illustrates that the introduction of TBS cross-timestep connections consistently outperforms models without such connections($n = 0$). This validates the efficacy of our cross-timestep linkage strategy based on the high-dimensional feature similarity of LDM. Among

the experiments incorporating cross-timestep connections, the models with 1 and 8 connections both achieved equally optimal results. The model with 1 connection demonstrated slightly superior performance in FID and Precision, whereas the model with 8 nodes exhibited marginally better outcomes in sFID and Recall.

Table 6: The number of TBS connections

| $n$ | FID↓ | sFID↓ | Prec.↑ | Recall↑ |
|---|---|---|---|---|
| 0 | 30.24 | 28.21 | 29.77 | 16.94 |
| 1 | 24.22 | 20.94 | 34.28 | 18.22 |
| 8 | 22.74 | 17.91 | 33.54 | 19.90 |
| 12 | 23.25 | 28.31 | 37.74 | 18.78 |

**Effects of SPD.** As a general quantization method, real-to-binary [39] suggests that using attention map-based loss during the distillation of a binary model from a full-precision model achieves better results. In contrast, BinaryDM, as the work most closely related to BiDM, directly points out that using L2 loss makes it difficult to align and optimize binary features with full-precision features. These studies indicate that the general L2 loss is inadequate for meeting the optimization needs of binary scenarios. So we also compare our SPD with the commonly used L2 loss function. As shown in Table 7, by replacing the L2 loss function with patch distillation, the model can achieve better performance.

Table 7: Different distillation strategies

| $\mathcal{L}_{distil}$ | FID↓ | sFID↓ | Prec.↑ | Recall↑ |
|---|---|---|---|---|
| $\mathcal{L}_2$ | 26.07 | 23.26 | 33.12 | 18.98 |
| $\mathcal{L}_{SPD}$ | **22.74** | **17.91** | **33.54** | **19.90** |

**Further Inference Efficiency Analysis.** We expand upon the inference process described in Eq.9 and provide a detailed explanation and testing. Since the divisor involved in calculating the mean of $A^{1,h,w}$ from $I^{c,h,w}$ (i.e., the channel dimension $c$) can be integrated into $k^{1,1,3,3}$ in advance, resulting in $k'^{1,1,3,3} = \frac{k^{1,1,3,3}}{c}$. Additionally, $\alpha^{n,1,1,1}$ derived from $W^{n,c,h,w}$ can also be computed ahead of inference. Therefore, the actual operations involved during inference are as follows:

[FP] Original full-precision convolution:

- (0) Perform convolution between full-precision $I_f^{c=448,h=32,w=32}$ and full-precision $W_f^{n=448,c=448,h=32,w=32}$ to obtain the full-precision output $O_f^{448,32,32}$.

[XNOR-Net/BiDM] The inference process for XNOR-Net/BiDM involves the following 6 steps:

- Sign operation:

    (1) Sign operation:

- Binary operation:

    (2) Perform convolution between the binary $I_b^{448,32,32}$ and the binary $W_b^{448,448,3,3}$ to obtain the full-precision output $O_f^{448,32,32}$.

- Full-precision operations:

    (3) Sum the full-precision $I_f^{448,32,32}$ across channels to obtain $A^{1,32,32}$.

    (4) Perform convolution between full-precision $A^{1,32,32}$ and $k'^{1,1,3,3}$ to obtain $O_1^{1,32,32}$.

    (5) Pointwise multiply $O_f^{448,32,32}$ by $O_1^{1,32,32}$ to obtain the full-precision output $O_2^{448,32,32}$.

    (6) Pointwise multiply $O_2^{448,32,32}$ by $\alpha^{448,1,1}$ to obtain the final full-precision output $O^{448,32,32}$.

We utilized the general deployment library Larq [8] on a Qualcomm Snapdragon 855 Plus to test the actual runtime efficiency of the aforementioned single convolution. The runtime results for a single inference are summarized in the Table 8. Due to limitations of the deployment library and hardware, Baseline achieved a 9.97x speedup, while XNOR-Net / BiDM achieved an 8.07x speedup. Besides, the improvement in generation performance brought by BiDM is even more significant, and we believe that it could achieve better acceleration results in a more optimized environment.

Table 8: The actual runtime efficiency of a single convolution.

| Method | (0) | (1)+(2) | (3) | (4) | (5) | (6) | Runtime($\mu s$ /convolution) | FID$\downarrow$ |
|---|---|---|---|---|---|---|---|---|
| FP | 176371.0 | | | | | | 176371.0 | 2.99 |
| Baseline (DoReFa) | | 17695.2 | | | | 4.3 | 17699.5 | 188.30 |
| XNOR-Net / BiDM | | 17695.2 | 2948.8 | 1133.3 | 83.2 | 4.3 | 21864.8 | 22.74 |

**Further Training Efficiency Analysis.** BiDM consists of two techniques: TBS and SPD. The time efficiency analysis during training is as follows: (1) TBS includes the learnable convolution of scaling factors (Eq.10) and the cross-time step connection (Eq.12). The increase in training time due to the convolution of trainable scaling factors is minimal, as the depth of the convolution for scaling factors is only 1, and the size of the trainable convolution kernel is only $3 \times 3$. The cross-time step connection is the primary factor for the increase in training time. Since it requires training $\alpha$, we introduce this structure during training, so each training sample requires not only noise estimation for $T^{t-1}$ but also for $T^t$, directly doubling the sampling steps. (2) SPD may lead to a slight increase in training time (an additional 0.18 times), but since we only apply supervision to the larger upsampling/middle/downsampling blocks, the increase is limited.

The results in Figure 6 align well with the theoretical analysis mentioned above. BiDM achieved significantly better generative results than baseline methods under the same training iterations, demonstrating that it not only has a higher upper limit of generative capability but is also relatively efficient when considering generative performance.

We also tested the FID after uniformly training for 0.5 days, and the results in Tabel 9 show: (1) BiDM has the best convergence, even in a short training time. (2) No.3 significantly outperforms No.5 because connections across timesteps greatly increase training time, making No.3 converge faster in the early training stages. (3) No.5 slightly outperforms No.7 because $\mathcal{L}_{SPD}$ causes a slight increase in training time.

We emphasize that the biggest challenge in fully binarizing DM lies in the drop in accuracy. Although BiDM requires a longer training time for the same number of iters, it significantly enhances the quality of generated images, as no other method has been able to produce effective images.

Table 9: Training speed under different settings, and FID at 0.5 days.

| No. | convolution of scaling factors (Eq.9) | learnable $k$ | connections across timesteps | $\mathcal{L}_{distil}$ | Training Speed (ms/iter) | FID$\downarrow$ at 0.5 days |
|---|---|---|---|---|---|---|
| 1 | | | | | 309.8 | 167.59 |
| 2 | $\checkmark$ | | | | 310.2 | 121.63 |
| 3 | $\checkmark$ | $\checkmark$ | | | 340.8 | 58.55 |
| 4 | $\checkmark$ | | $\checkmark$ | | 458.5 | 93.66 |
| 5 | $\checkmark$ | $\checkmark$ | $\checkmark$ | | 480.2 | 70.80 |
| 6 | $\checkmark$ | | | $\mathcal{L}_{SPD}$ | 389.6 | 86.78 |
| 7 | $\checkmark$ | $\checkmark$ | $\checkmark$ | $\mathcal{L}_2$ (MSE) | 496.8 | 71.15 |
| 8 | $\checkmark$ | $\checkmark$ | $\checkmark$ | $\mathcal{L}_{SPD}$ | 547.2 | 47.11 |

# C  Additional Visualization Results

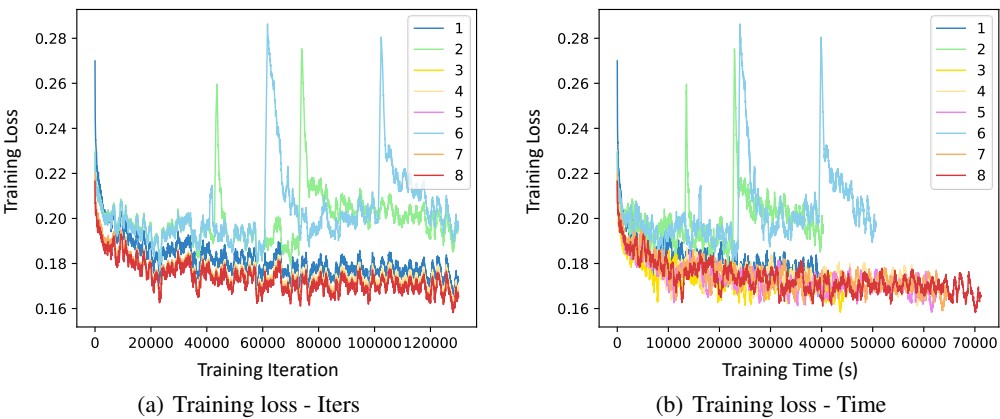

|     |     |
|:---:|:---:|
| (a) Training loss - Iters | (b) Training loss - Time |

Figure 6: (a) Training iterations and training loss under different settings. (b) Training time and training loss under different settings. The meaning of the numbers in the legend corresponds to those in Table 9.

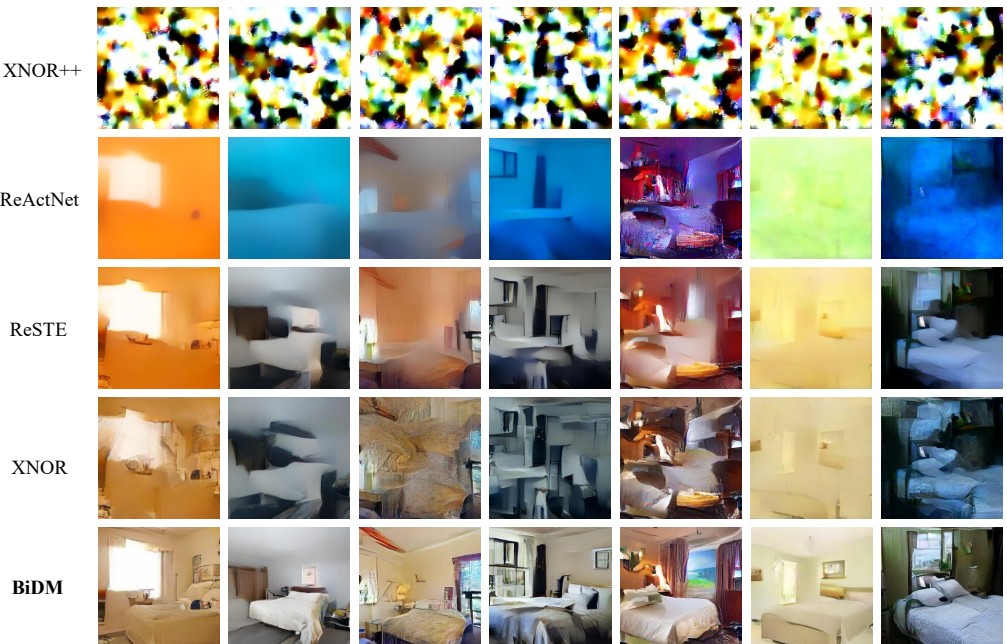

Figure 7: Generation results of BiDM and baselines on the LSUN-Bedrooms dataset.

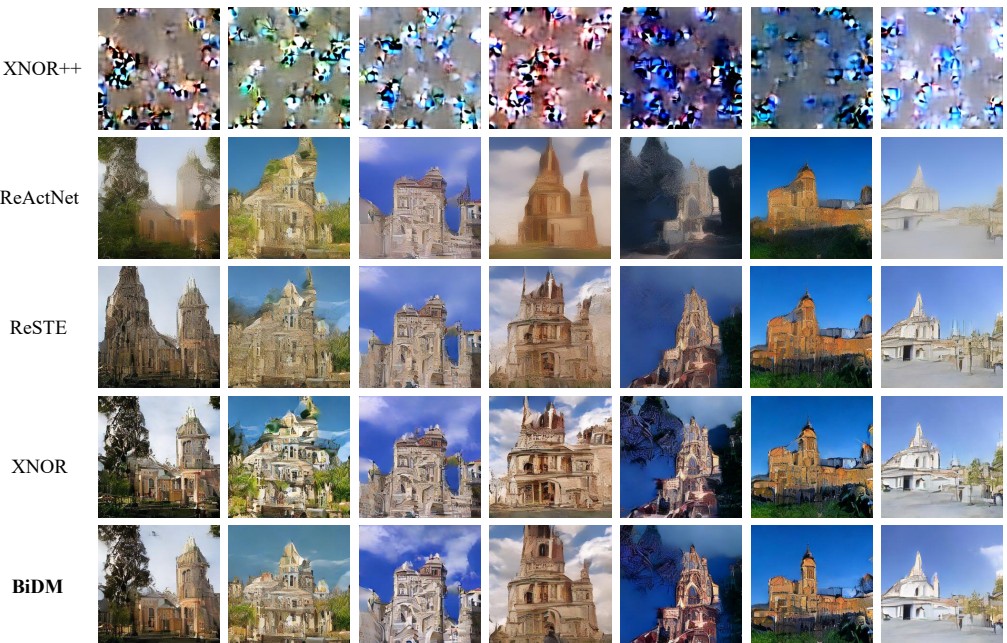

Figure 8: Generation results of BiDM and baselines on the LSUN-Churches dataset.

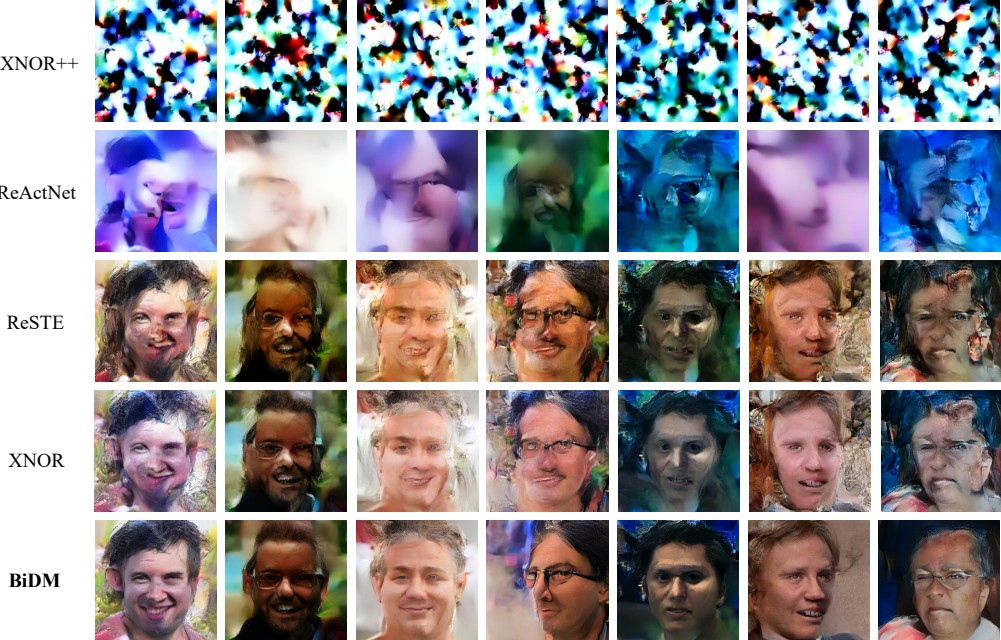

Figure 9: Generation results of BiDM and baselines on the FFHQ dataset.

