# OpenReview forum: "BiDM: Pushing the Limit of Quantization for Diffusion Models"
_NeurIPS.cc/2024/Conference — NeurIPS 2024 poster_

### Official Review · Reviewer_AtKg · 2024-06-27

**Soundness:** 2
**Presentation:** 3
**Contribution:** 2
**Rating:** 5
**Confidence:** 4

**Summary:**

This paper proposes BiDM, which focuses on quantizing both weights and activations of diffusion models (DMs). Specifically, the authors introduce:
- Cross-timestep feature connection to enhance the accuracy of noise prediction in binarized DMs.
- Space-patched distillation, a novel variant of distillation loss that emphasizes spatial locality.

**Strengths:**

- Binarization of diffusion models (DMs) represents a promising research area aimed at accelerating sampling processes, and this work appears to be the first to fully binarize DMs to W1A1 to the best of my knowledge.

- The introduction of cross-timestep information connection is innovative in enhancing the performance of binarized DMs.

- The experimental results on LSUN-Bedrooms, LSUN-Churches, and FFHQ effectively demonstrate the effectiveness of BiDM and XNOR techniques.

**Weaknesses:**

- The experimental results lack sufficient conviction. Tables 1 and 2 compare BiDM primarily with works focused on quantizing discriminative models, which may not be the most appropriate choice of baseline. It would be more informative to include comparisons with quantization methods designed specifically for diffusion models, such as EfficientDM [cite], given its open-sourced nature and compatibility under W1A1 settings. This comparison is crucial for a comprehensive evaluation.

- Figure.4 may not effectively demonstrate the advantages of paying attention to local information. To better illustrate the effectiveness of Space-Patched Distillation (SPD), the authors should compare experimental results with those obtained using vanilla distillation loss (Mean Squared Error between outputs of a full precision model and a binarized model).

- The proposed binarization method introduces additional convolution computations between two matrices in floating-point, which may not be efficient for hardware deployment. Moreover, the dynamic calculation of matrix $A$ during inference introduces memory access overhead that could be significant in practical deployment scenarios.

- During inference, matrix multiplications (MMs) in the attention mechanism typically consume considerable time. It appears that this work retains these MMs in floating-point arithmetic, which could impact efficiency.

**minor issues**
- Line 93: The citation for BinaryDM is incorrectly linked.
- In Figure 4, the caption states "Ours denotes the self-attention," but in the figure, it's labeled as "SPD."
- The term "self-attention" in Section 3.3 is confusing; perhaps using an alternative term would be clearer.

**Questions:**

- Are the Q, K, V in the attention modules binarized during inference?

- During inference, is $A$ in Equation 9 calculated from the inputs?

- In Line 224, the authors claim that "conventional $l_2$  loss makes it difficult for binary DMs to achieve direct matching." Could you provide evidence or reference existing works that have verified this claim?

- Figure 3 is somewhat confusing. Why is $L^{t-1}$ a linear scaling up of $B^{t-1}$? Is this relationship depicted by Equation 12?

**Limitations:**

This paper includes a discussion of limitations.

---

> ### Author Rebuttal · Authors · 2024-08-07
>
> Thank you for your detailed review of our work. Here are our responses to your concerns:
>
> > Q1: The experimental ...
>
> Thank you for your suggestion. We have added experiments and discussions on EfficientDM. You can check the Global Rebuttal (1) for more details.
>
>
>
> > Q2: Figure.4 ...
>
> Thank you for your suggestion. We have already placed this exploration in the manuscript. Table.7 shows that $L_2$ represents the results using MSE Loss as the distillation loss. $L_{SPD}$, with an FID of 22.74, significantly outperforms MSE Loss, which has an FID of 26.07, demonstrating the suitability of SPD for DM.
>
>
>
> > Q3: The proposed ...
>
> It should be noted that during inference, BiDM requires only a very small number of additional floating-point additions for the connections across timesteps compared to the classic binarization work XNOR-Net, and there are no differences in the majority of calculations, such as convolutions. XNOR-Net's original paper already stated that it achieves a 58× speedup in convolution operations in its abstract.
>
> We understand your concerns about efficiency, but performing a floating-point convolution with a depth of 1 for scaling factors requires only a very small amount of computation, and the overhead for averaging matrix $A$ is also minimal. Statistical results show that this portion accounts for only 0.08% of the total OPs in binarized DM. Therefore, BiDM remains efficient during inference.
>
>
>
> > Q4: During inference ...
>
> BiDM fully binarizes components such as convolutions and QKV, and the remaining components have a very minimal impact on computation. Specifically, the computational cost of this part accounts for only 0.38% of the total OPs in the binarized DM.
>
> In particular, we fully binarized linear/conv1d layers with weights and MMs involving the input x in AttentionBlock, while MMs involving intermediate results (such as proj_out) were performed using floating-point operations. This decision was based on our observation that the latter does not involve weight storage, involves very little computation, and has a relatively noticeable impact on the final results.
>
> Thus, we chose not to fully binarize the mentioned MMs, and this trade-off is necessary to achieve a more accurate fully binarized DM.
>
>
>
> > Q5: minor issues:
> >
> > Q5-1: Line 93 ...
>
> Sorry for the incorrect link; it should have pointed to reference [54] in the manuscript.
>
> > Q5-2: In Figure 4 ...
>
> Here, we provide a clearer explanation to present the meaning of Fig.4 better:
>
> - The first row labeled `FP32` shows the visualization of the full-precision model output $\mathcal{F}^{fp}$.
> - The second row labeled `Diff` represents the visualization of the difference between the full-precision model output and the binarized model output, $\mathcal{F}^{fp} - \mathcal{F}^{bi}$.
> - The third row labeled `SPD` shows the visualization of the local attention for both the full-precision model and the binarized model calculated using the SPD method: $ \frac{ A^{fp}}{ ||A^{fp}||_2 } - \frac{A^{bi}}{ ||A^{bi}||_2 } $.
>
> From Figure 4, it can be observed that our SPD method allows the model to pay more attention to local information in each patch.
>
> > Q5-3: The term ...
>
> Thank you for your feedback. Using "attention-guided" might be a more appropriate term. Our implementation of SPD is inspired by self-attention, involving matrix multiplication with its transpose locally to measure local self-attention. However, as you mentioned, self-attention is a widely accepted technical term, and using it to describe our SPD could lead to conceptual confusion.
>
> We will update the explanations for these three issues in the revised version of the manuscript. Thank you very much for your detailed review of our work.
>
>
>
> > Q6: Are the Q ...
>
> Sure, as we mentioned in our response to `Q4` above, QKV components have all been binarized.
>
>
>
> > Q7: During inference ...
>
> Yes, the calculations are dynamic. However, as explained in `Q3`, this part of the computation is identical to that in XNOR-Net and involves very minimal additional calculation, resulting in a very limited extra burden on the inference process.
>
>
>
> > Q8: In Line 224 ...
>
> As a general quantization method, real-to-binary [1] suggests that using attention map-based loss during the distillation of a binary model from a full-precision model achieves better results. In contrast, BinaryDM, as the work most closely related to BiDM, directly points out that using L2 loss makes it difficult to align and optimize binary features with full-precision features. These studies indicate that the general L2 loss is inadequate for meeting the optimization needs of binary scenarios.
>
> As a result, Table.7 of our manuscript shows that SPD indeed outperforms L2 loss with MSE.
>
> We will include a discussion on this section in the revised version of the manuscript to make the motivation behind SPD more clear.
>
> [1] Training binary neural networks with real-to-binary convolutions.
>
>
>
> > Q9: Figure 3 ...
> >
> > (1) Why is Lt−1 ...
> >
> > (2) Is this relationship ...
>
> Sorry for the confusion. Here are our explanations for the two issues:
>
> - (1) In reality, although the trainable $k$ results in changes to local elements, this does not imply that the entire vector is linearly scaled. Thus, ${L}^{t-1}$ is not a linear scaling up of ${B}^{t-1}$.
> - (2) The relationship between these two is unrelated to Equation 12. ${L}^{t-1}$ and ${B}^{t-1}$ should be considered as differences in the output features of the model obtained after training under non-trainable/trainable settings of $k$ in Eq.10. This means that the learnable $k$ allows for more flexible and free feature representation.
>
> You could refer to the Global Rebuttal (2) for a further explanation.

---

> ### Author Response · Authors · 2024-08-12
>
> Dear Reviewer AtKg,
>
> Thank you for your thorough review of our work, BiDM, during the review stage. We have carefully considered your concerns during the rebuttal stage and made revisions to the relevant sections of the manuscript.
>
> We are looking forward you to reviewing our response and we are also willing to answer any further questions.
>
> Best regards,
>
> Authors

---

> ### Comment · Reviewer_AtKg · 2024-08-13
> **Thanks for the responses.**
>
> Thanks for the authors' response. My concerns regarding **Q1, Q2, Q4, Q5, Q6, Q8** have been satisfactorily addressed. Nonetheless, I remain uncertain about the deployment efficiency, particularly given that practical inference speeds often diverge from theoretical operations per second (OPs). The dynamic *min-max* operations, despite their minimal theoretical OPs, may introduce substantial latency in practical deployment, especially with smaller model sizes where matrix multiplications are less significant. I am keen to know if there are effective tools available to verify this.
>
> Regarding **Q3**, could you elaborate on the computations in Eq.(9)? Specifically, how is the convolution between $A$ and $ k\alpha $ executed efficiently?

---

> > ### Author Response · Authors · 2024-08-13
> > **Re: Further response (2/2)**
> >
> > > Q11: Regarding **Q3**, could you elaborate on the computations in Eq.(9)? Specifically, how is the convolution between A and kα executed efficiently?
> >
> > Yes, in our response to `Q10`, we provided a detailed breakdown of the computation sequence in Eq.(9), including the pre-computed $k'$ and $\alpha$, and the calculations (1)~(6) that need to be performed during inference.
> >
> > For the convolution between $ A $ and $ k\alpha $, the following calculations are involved:
> >
> > - (a) Pre-computation before inference, which does not need to be repeated during inference. This includes:
> >   - The divisor $ c $ (the channel dimension) involved in calculating the mean of $ A^{1,h,w} $ from $ I^{c,h,w} $ in a single convolution
> >   - $ \alpha^{n,1,1,1} $ obtained from $ W^{n,c,h,w} $
> >   - $ k'^{1,1,3,3} = \frac{k^{1,1,3,3}}{c} $
> > - (b) As mentioned in `Q10` (4):
> >   - Perform the convolution between full-precision $ A^{1,32,32} $ and $ k'^{1,1,3,3} $ to obtain $ O_1^{1,32,32} $
> > - (c) $ \alpha $ needs to be used in the operation in `Q10` (6): $ O_2^{448,32,32} \odot \alpha^{448,1,1,1} $
> >
> > The efficiency of our operators (as in XNOR-Net) is reflected in the following:
> >
> > - The operations in (a) can be pre-computed before inference, which means:
> >   - Step (3) in `Q10` does not involve high-cost multiplication or division operations.
> >   - Step (6) in `Q10` does not require computing $ \alpha $ during inference.
> > - The convolution performed in part (b) has a channel count of only 1.
> > - Step (5) and (6) in `Q10` are point-wise multiplications rather than matrix multiplications, resulting in a lower computational burden.
> >
> > In terms of actual inference time, the inference efficiency of our operators is only 0.19x slower than that of a fully binarized convolution in a Baseline without any full-precision scaling factors, even though we use full-precision inference components, consistent with XNOR-Net, to achieve viewable generative performance.

---

> > > ### Comment · Reviewer_AtKg · 2024-08-13
> > > **Thanks for the Response**
> > >
> > > My concerns are well addressed and I would like to raise my score by 1 point. Thanks for the authors' detailed responses.

---

> ### Author Response · Authors · 2024-08-13
> **Re: Further response (1/2)**
>
> Thank you for your prompt response. We are glad to clarify the inference efficiency of BiDM.
>
> > Q10: Thank you for the authors' response. My concerns regarding **Q1, Q2, Q4, Q5, Q6, Q8** have been satisfactorily addressed. Nonetheless, I remain uncertain about the deployment efficiency, particularly given that practical inference speeds often diverge from theoretical operations per second (OPs). The dynamic *min-max* operations, despite their minimal theoretical OPs, may introduce substantial latency in practical deployment, especially with smaller model sizes where matrix multiplications are less significant. I am keen to know if there are effective tools available to verify this.
>
> In BiDM, there are no dynamic *min-max* operations; instead, our design involves only basic arithmetic operations such as convolution, matrix multiplication, addition and summation, and matrix dot multiplication. Additionally, as stated in Eq.(9) of our manuscript and in our response to `Q3`, the operators in BiDM behave the same as those in XNOR-Net[1] during inference, which corresponds to Eq.(11) in the XNOR-Net's original paper.
>
> Following your suggestion, we expand upon the inference process described in `Eq.(9)` and provide a detailed explanation and testing. Since the divisor involved in calculating the mean of $ A^{1,h,w} $ from $ I^{c,h,w} $ (i.e., the channel dimension $ c $) can be integrated into $ k^{1,1,3,3} $ in advance, resulting in $ k'^{1,1,3,3} = \frac{k^{1,1,3,3}}{c} $. Additionally, $ \alpha^{n,1,1,1} $ derived from $ W^{n,c,h,w} $ can also be computed ahead of inference. Therefore, the actual operations involved during inference are as follows:
>
> [FP] Original full-precision convolution:
>
> - (0) Perform convolution between full-precision $I_f^{c=448,w=32,h=32}$ and full-precision $W_f^{n=448,c=448,w=32,h=32}$ to obtain the full-precision output $O_f^{448,32,32}$.
>
> [XNOR-Net/BiDM] The inference process for XNOR-Net/BiDM involves the following 6 steps:
>
> - Sign operation:
>   - (1) Compute $I_b^{448,32,32} = \text{sign}(I_f^{448,32,32})$.
> - Binary operation:
>   - (2) Perform convolution between the binary $I_b^{448,32,32}$ and the binary $W_b^{448,448,3,3}$ to obtain the full-precision output $O_f^{448,32,32}$.
> - Full-precision operations:
>   - (3) Sum the full-precision $I_f^{448,32,32}$ across channels to obtain $A^{1,32,32}$.
>   - (4) Perform convolution between full-precision $A^{1,32,32}$ and $k'^{1,1,3,3}$ to obtain $O_1^{1,32,32}$.
>   - (5) Pointwise multiply $O_f^{448,32,32}$ by $O_1^{1,32,32}$ to obtain the full-precision output $O_2^{448,32,32}$
>   - (6) Pointwise multiply $O_2^{448,32,32}$ by $\alpha^{448,1,1}$ to obtain the final full-precision output $O^{448,32,32}$
>
> We utilized the general deployment library Larq[2] on a Qualcomm Snapdragon 855 Plus to test the actual runtime efficiency of the aforementioned single convolution. The runtime results for a single inference are summarized in the table below. Due to limitations of the deployment library and hardware, Baseline achieved a 9.97x speedup, while XNOR-Net/BiDM achieved an 8.07x speedup. Besides, the improvement in generation performance brought by BiDM is even more significant, and we believe that it could achieve better acceleration results in a more optimized environment.
>
> |                  |   (0)    | (1)+(2) |  (3)   |  (4)   | (5)  | (6)  | Runtime($\mu s$/convolution) | FID$\downarrow$ |
> | :--------------: | :------: | :-----: | :----: | :----: | :--: | :--: | ---------------------------: | --------------: |
> |        FP        | 176371.0 |         |        |        |      |      |                     176371.0 |            2.99 |
> | Baseline(DoReFa) |          | 17695.2 |        |        |      | 4.3  |                      17699.5 |          188.30 |
> | XNOR-Net / BiDM  |          | 17695.2 | 2948.8 | 1133.3 | 83.2 | 4.3  |                      21864.8 |           22.74 |
>
> [1] Rastegari et al. Xnor-net: Imagenet classification using binary convolutional neural networks. ECCV 2016
>
> [2] "LarQ". https://github.com/larq/larq

---

### Official Review · Reviewer_cNoX · 2024-07-09

**Soundness:** 3
**Presentation:** 3
**Contribution:** 3
**Rating:** 7
**Confidence:** 4

**Summary:**

The manuscript proposes a method for fully binarizing both the weights and activations of diffusion models, named BiDM. Structurally, it introduces an improved XNOR method for scaling factors of activations and high-level feature connections across time steps, based on observations of existing temporal phenomena. For optimization, the manuscript presents a patch-based attention mechanism distillation method, approached from the perspective of spatial features. Quantitative analysis and generated examples demonstrate that BiDM, as the first work to fully binarize diffusion models, surpasses existing binarization methods, showcasing the potential for deploying diffusion models in resource-constrained scenarios.

**Strengths:**

1.	This is the first work to fully binarize diffusion models, preventing them from collapsing in extreme scenarios and demonstrating acceptable generated samples. This is significant for exploring the compression potential of generative models.
2.	The improvement to XNOR is not only applicable to diffusion models (DMs) but also has positive implications for the whole binarization field. Most papers and deployment frameworks, when replicating and implementing XNOR, have consistently followed the approach described in Equations 7 or 8 of this manuscript, rather than the method described in Equation 9. The authors have meticulously examined and adaptively improved the original XNOR approach within the DM field, prompting reconsideration in both DM compression and general binarization fields.
3.	The cross-time step connection appears novel and well-suited to the inference structure of DMs. Unlike the cache design in DeepCache, which aims at inference efficiency, this manuscript uses connections to enhance information due to the inherently efficient nature of binarized models. These connections can be placed at multiple nodes within the model, with learnable coefficient factors providing greater adaptability.
4.	Section 3.2 first designs TBS from the perspective of information enhancement and then provides further explanation from the perspective of error reduction in Figure 3, making the overall method design appear cohesive and intuitively clear.
5.	The distillation strategy design is straightforward and effective, approaching patch division based on the inherent requirements of the generation task and the natural locality of the convolution module, and achieving excellent results.

**Weaknesses:**

1. The description of the cross-time step connection during the training phase is somewhat vague. During the inference phase, the impact of this connection is iteratively attenuated. For example, when α is 0.3, the influence of step T on step T-2 is 0.3×0.3=0.09, rather than 0. However, upon reviewing the source code in the supplementary materials, I found that the authors only consider steps T-1 and T-2 during training. I would like to know if this affects the final accuracy.
2. The scenario assumed in Figure 3 is overly idealized. When only L^t changes, for instance, when L^t is at the lower right of L^(t-1), the weighted average might result in T^(t-1) being further from F^(t-1). How is this situation explained?
3. Although BiDM shows significant improvements over other binarization methods, there remains a substantial performance gap compared to full-precision models. This could still hinder its practical application.

**Questions:**

See Weaknesses

**Limitations:**

The manuscript only addresses diffusion models using a U-Net backbone primarily based on convolution. It appears to be inapplicable to diffusion models like DiT, which are primarily based on transformers and not related to U-Net.

---

> ### Author Rebuttal · Authors · 2024-08-07
>
> Thank you very much for your high recognition of our work and the valuable suggestions you provided. Our response is as follows:
>
> > Q1: The description of the cross-time step connection during the training phase is somewhat vague. During the inference phase, the impact of this connection is iteratively attenuated. For example, when α is 0.3, the influence of step T on step T-2 is 0.3×0.3=0.09, rather than 0. However, upon reviewing the source code in the supplementary materials, I found that the authors only consider steps T-1 and T-2 during training. I would like to know if this affects the final accuracy.
>
> From the optimization principles of DDPM[1], this does not affect the accuracy of the results. Essentially, our training process is no different from that of DDPM, as both use efficient training to optimize random terms of the usual variational bound on negative log likelihood with stochastic gradient descent. Due to the cross-time step connection in TBS, which requires considering the values from the previous inference step, we included T-2, T-1, and T in our considerations. This approach still adheres to the DDPM training methodology.
>
> [1] Ho J, Jain A, Abbeel P. Denoising diffusion probabilistic models. NeurIPS, 2020.
>
>
>
> > Q2: The scenario assumed in Figure 3 is overly idealized. When only L^t changes, for instance, when L^t is at the lower right of L^(t-1), the weighted average might result in T^(t-1) being further from F^(t-1). How is this situation explained?
>
> You can refer to the Global Rebuttal (2) for the explanation. Additionally, in practical applications, the learnability of $\alpha$ ensures the effective collaboration between $L^{t-1}$ and $L^{t}$. Moreover, the ablation study results in Table.6 also demonstrate the effectiveness of connecting $L^{t-1}$ and $L^{t}$ across time steps.
>
>
>
>
>
> > Q3: Although BiDM shows significant improvements over other binarization methods, there remains a substantial performance gap compared to full-precision models. This could still hinder its practical application.
>
> BiDM is the first fully binarized DM method capable of generating viewable images, significantly surpassing advanced binarization methods like ReSTE in quantitative metrics (on LSUN-Bedrooms, BiDM achieves an FID of 22.74, notably better than ReSTE's FID of 59.44). This demonstrates the feasibility of fully binarized DMs, marking a step towards practical applications with great potential.
>
> Besides, as you highlighted in the Strengths section, the improvements to techniques like XNOR will have a positive impact on the entire binarization field. We will further explore this area in future work to achieve broader practical applications.
>
>
>
> > Q4: The manuscript only addresses diffusion models using a U-Net backbone primarily based on convolution. It appears to be inapplicable to diffusion models like DiT, which are primarily based on transformers and not related to U-Net.
>
> The connections across timesteps in TBS and SPD should be directly applicable to DiT, with SPD likely being even more compatible with DiT's inherently space-patched input. While the convolutional design in TBS isn't suitable for DiT's linear-based architecture, we have observed that finer quantization granularity (such as per-group quantization) is gradually being proposed for architectures like transformers. Using a similar approach of dynamically calculating statistics for scaling factors followed by convolution or linear transformation could also be applicable. We plan to explore these aspects further in our future work.

---

> ### Author Response · Authors · 2024-08-12
>
> Dear Reviewer cNoX,
>
> Thank you for your thorough review of our work, BiDM, during the review stage. We have carefully considered your concerns during the rebuttal stage and made revisions to the relevant sections of the manuscript.
>
> We are looking forward you to reviewing our response and we are also willing to answer any further questions.
>
> Best regards,
>
> Authors

---

> > ### Comment · Reviewer_cNoX · 2024-08-13
> >
> > Thank you for the detailed rebuttal and the additional results provided. I appreciate the effort in addressing the issues raised. Due to the novelty and potential broad applications, I am willing to vote for acceptance for this paper.

---

### Official Review · Reviewer_4HDt · 2024-07-11

**Soundness:** 2
**Presentation:** 2
**Contribution:** 2
**Rating:** 5
**Confidence:** 3

**Summary:**

This paper aims to fully binarize weights and activations of diffusion models (DMs) to achieve storage saving and inference acceleration. To this end, the paper proposes timestep-friendly binary structure (TBS), which employs learnable activation binarizers and cross-timestep feature connections to capture the correlation of the activation features over the timesteps. In addition, the paper introduces space patched distillation (SPD) to match the spatial locality of binary features with the full-precision ones during training. The method is tested on some common image generation benchmarks such as LSUN and FFHQ 256x256, CIFAR-10.

**Strengths:**

- The paper aims to tackle a challenging problem which is to fully binarize both weights and activations of diffusion models.
- The paper is well-written.
- The experimental results are promising.
- The proposed methods of using timestep-friendly binary structure and space patched distillation are well-motivated.

**Weaknesses:**

- The main weakness of the paper is that the proposed method, BiDM, increases the training time of DMS compared to the original process. This issue should be more precise in the paper. The authors should compare the convergence rates of the proposed method and its competitors in terms of wall-clock time and identify which components contribute to the increased training time by breaking down the complexity of all main components of the proposed model.
- The chosen baselines are quite weak as all of them were originally designed for discriminative tasks such as image classification. The authors should consider stronger baselines dedicated to generative tasks such as [1] and the baselines therein.

Ref:

[1] Xia et al. Basic Binary Convolution Unit for Binarized Image Restoration Network. ICLR 2023

**Questions:**

In Table 3, it is not clear why adding SPD alone makes the results worse, whereas combining it with TBS leads to improved performance. Do the authors have any explanation for this phenomenon? This point should be addressed to clarify the underlying reasons for the observed behavior and to provide a deeper understanding of the interaction between SPD and TBS in the proposed method.

**Limitations:**

Yes

---

> ### Author Rebuttal · Authors · 2024-08-07
>
> Thank you for reviewing our manuscript and providing valuable suggestions. Here are our responses to some of the concerns you raised:
>
> > Q1: The main weakness of the paper is that the proposed method, BiDM, increases the training time of DMS compared to the original process. This issue should be more precise in the paper. The authors should compare the convergence rates of the proposed method and its competitors in terms of wall-clock time and identify which components contribute to the increased training time by breaking down the complexity of all main components of the proposed model.
>
> Thank you for your suggestions. BiDM consists of two techniques: TBS and SPD. The time efficiency analysis during training is as follows:
>
> - TBS includes the learnable convolution of scaling factors (Eq.10) and the cross-time step connection (Eq.12):
>   - The increase in training time due to the convolution of trainable scaling factors is minimal, as the depth of the convolution for scaling factors is only 1, and the size of the trainable convolution kernel is only $3\times3$.
>   - The cross-time step connection is the primary factor for the increase in training time. Since it requires training $\alpha$, we introduce this structure during training, so each training sample requires not only noise estimation for $T^{t-1}$ but also for $T^{t}$, directly doubling the sampling steps.
>
> - SPD may lead to a slight increase in training time (an additional 0.18 times), but since we only apply supervision to the larger upsampling/middle/downsampling blocks, the increase is limited.
>
> We have supplemented the actual training time on an NVIDIA A100 40GB GPU, and the results in Global Rebuttal (3) align well with the theoretical analysis mentioned above. Due to the actual software and hardware frameworks, the actual training time per iter for BiDM did not completely double compared to the baseline.
>
> Following your suggestion, we compared the training loss under the same training iterations or training time. BiDM achieved significantly better generative results than baseline methods under the same training cost, demonstrating that it not only has a higher upper limit of generative capability, but is also relatively efficient when considering generative performance. You can refer to Global Rebuttal (3) for a more detailed explanation.
>
> We also tested the FID after uniformly training for 0.5 days, and the results in Global Rebuttal (3) show:
>
> - BiDM has the best convergence, even in a short training time.
> - No.3 significantly outperforms No.5 because connections across timesteps greatly increase training time, making No.3 converge faster in the early training stages.
> - No.5 slightly outperforms No.7 because $\mathcal{L}_{SPD}$ causes a slight increase in training time.
>
> We emphasize that the biggest challenge in fully binarizing DM lies in the drop in accuracy. Although BiDM requires a longer training time for the same number of iters, it significantly enhances the quality of generated images, as no other method has been able to produce effective images.
>
>
>
> > Q2: The chosen baselines are quite weak as all of them were originally designed for discriminative tasks such as image classification. The authors should consider stronger baselines dedicated to generative tasks such as [1] and the baselines therein.
>
> Thank you for your suggestion. We have supplemented the results and analysis for BBCU, and we have also included experimental results for other quantization methods suited for generative models, such as EfficientDM. You can refer to the Global Rebuttal (1) for more comprehensive information.
>
> We will include the above discussion in the revised version of the manuscript. These discussions will further clarify the motivation and necessity behind each component of BiDM.
>
>
>
> > Q3: In Table 3, it is not clear why adding SPD alone makes the results worse, whereas combining it with TBS leads to improved performance. Do the authors have any explanation for this phenomenon? This point should be addressed to clarify the underlying reasons for the observed behavior and to provide a deeper understanding of the interaction between SPD and TBS in the proposed method.
>
> Sorry for the confusion, but there might be a misunderstanding. In fact, SPD alone improves the results compared to Vanilla. Specifically, the ablation results in Table 3 can be more clearly restated in the table below. In all metrics, the binarized DM with only SPD outperforms the Vanilla method. For example, FID decreases from 106.62 to 40.62, demonstrating its superiority in optimizing binarized DMs.
>
> When SPD and TBS are used together, SPD brings even better generative performance on top of TBS, further reducing the FID from 35.23 to 22.74.
>
> | Method  |   TBS    |   SPD    | FID$\downarrow$ | sFID$\downarrow$ | Prec.$\uparrow$ | Recall$\uparrow$ |
> | ------- | :------: | :------: | --------------: | ---------------: | --------------: | ---------------: |
> | Vanilla |          |          |          106.62 |            56.81 |            6.82 |             5.22 |
> | +TBS    | $\surd $ |          |           35.23 |            25.13 |           26.38 |            14.32 |
> | +SPD    |          | $\surd $ |           40.62 |            31.61 |           23.87 |            11.18 |
> | BiDM    | $\surd $ | $\surd $ |           22.74 |            17.91 |           33.54 |            19.90 |

---

> > ### Comment · Reviewer_4HDt · 2024-08-12
> > **Official comment from Reviewer 4HDt**
> >
> > Thanks the author(s) for the rebuttal.
> >
> > As most of my concerns have been addressed, I will increase score by 1 point.

---

> ### Author Response · Authors · 2024-08-12
>
> Dear Reviewer 4HDt,
>
> Thank you for your thorough review of our work, BiDM, during the review stage. We have carefully considered your concerns during the rebuttal stage and made revisions to the relevant sections of the manuscript.
>
> We are looking forward you to reviewing our response and we are also willing to answer any further questions.
>
> Best regards,
>
> Authors

---

### Author Rebuttal · Authors · 2024-08-07

## Global Rebuttal

We appreciate all reviewers for their careful reviews and the constructive feedback provided on our work, BiDM. Here is a summary of the main contributions of BiDM:

We propose BiDM, the first method to achieve an accurate fully binarized diffusion model, aiming for extreme compression and inference acceleration. Based on two observations — activations at different timesteps and the characteristics of image generation tasks — we introduce the Timestep-friendly Binary Structure (TBS) and Space Patched Distillation (SPD) from temporal and spatial perspectives, respectively. In TBS, the learnable tiny convolution adapts to the highly dynamic activation range of DMs from the basic binary operator level. It is tightly integrated with cross-timestep connections that leverage the similarity of activation features between adjacent timesteps, forming the structure of BiDM. SPD takes advantage of the spatial locality in both the DM model structure and the image generation tasks it performs, alleviating optimization difficulties brought by general training methods or naive L2 distillation loss and achieving better generative results.

As the first fully binarized DM, BiDM achieves the best generative results, reducing FID by 62% on LSUN-Bedrooms and is currently the only method capable of generating visually acceptable images. At the same time, BiDM ensures considerable efficiency during inference. Compared to the classic binarization method XNOR-Net, it only adds a minimal amount of addition operations for cross-timestep connections, achieving 28.0× storage and 52.7× OPs savings.

We also noticed that the reviewers raised some common questions. We have summarized and responded to them collectively as follows:

(1) Reviewers suggested adding quantization methods more suited to generative models, such as BBCU and EfficientDM, for comparison. We have supplemented this on LSUN-Bedrooms and conducted the following analysis:

- BBCU:
  - We have supplemented our work with BBCU, a binarization method more akin to generative models like DMs rather than discriminative models. We implemented the residual connections in BBCU and used RPReLU. However, since DMs do not have BN layers, we did not incorporate the BN design from BBCU in our adaptation.
  - Experimental results indicate that even as a binarization strategy for generative models, BBCU faces significant breakdowns when applied to DMs.
- EfficientDM:
  - As a work targeting QAT for DM, EfficientDM is indeed a suitable comparison, especially since it designs TALSQ to address the variation in activation range. We adapted DoReFa as the basic operator for its use under W1A1.
  - The results in the table below show that EfficientDM struggles to adapt to the extreme scenario of W1A1, and this may be due to its quantizer having difficulty adapting to binarized DM, and using QALoRA for weight updates might yield suboptimal results compared to full-parameter QAT.

As we mentioned in the TBS section of our manuscript, most existing binarization methods struggle to handle the wide activation range and flexible expression of DMs, further highlighting the necessity of TBS. Their optimization strategies may also not be tailored for the image generation tasks performed by DM, which means they only achieve conventional but suboptimal optimization.

| Method      | #Bits | FID$\downarrow$ | sFID$\downarrow$ |
| :---------- | ----- | --------------: | ---------------: |
| BBCU        | 1/1   |          236.07 |            89.66 |
| EfficientDM | 1/1   |          194.45 |           113.24 |
| BiDM        | 1/1   |           22.74 |            17.91 |

(2) Some reviewers expressed confusion regarding the details of Fig.3. Here is a clearer clarification for Fig.3:

- Since the feature space is very high-dimensional, we could only illustrate it using schematic diagrams. The functions of the two components in TBS are exaggerated for a clearer explanation. Thus, Fig.3 is merely a visual representation of the collaborative function of the two components in TBS. For a detailed explanation, please refer to the end of section 3.2 in our manuscript. In simple terms, the learnable tiny convolution $k$ in TBS allows scaling factors to adapt more flexibly to the representation of DM, while connections across timesteps enable the binarized DM to use the previous step’s output information for appropriate information compensation. Together, these elements enhance the precision of the binarized DM.
- We have adjusted Fig.3 to make it more general. You can view the updated illustration in the attached PDF.

(3) We have also included in the PDF attachment the convergence of training loss over iterations/time for different methods. The results show that BiDM not only achieves the best generative performance with sufficient training time, as stated in the original manuscript but also exhibits the best convergence even under the same number of iterations/time.

We will include the above discussions in the revised version of the manuscript. These discussions will further clarify the motivation and necessity behind each component of BiDM.

For specific questions raised by each reviewer, please refer to our corresponding responses.

---

### Comment · Area_Chair_MBb7 · 2024-08-10
**Rebuttal Discussion**

Dear Reviewers,

The authors have provided a response to the comments. Please respond to the rebuttal actively.

Best,
AC

---

### Decision · Program_Chairs · 2024-09-25

**Decision:**

Accept (poster)

**Comment:**

This paper presents BiDM, a method for fully binarizing both the weights and activations of diffusion models. Techincally, BiDM consists of a timestep-friendly binary structure (TBS) to capture the correlation of the activation features over the timesteps and a space patched distillation (SPD) to match the spatial locality of binary features with the full-precision ones during training. Results on common image generation benchmarks were reported. The initial concerns were addressed during rebuttal. I recommend to accept, and encourage the authors to incorporate the discussions into the final version.